# Unraveling the Complex Interconnection between Specific Inflammatory Signaling Pathways and Mechanisms Involved in HIV-Associated Colorectal Oncogenesis

**DOI:** 10.3390/cancers15030748

**Published:** 2023-01-25

**Authors:** Botle Precious Damane, Thanyani Victor Mulaudzi, Sayed Shakeel Kader, Pragalathan Naidoo, Suzana D. Savkovic, Zodwa Dlamini, Zilungile Lynette Mkhize-Kwitshana

**Affiliations:** 1Department of Surgery, Steve Biko Academic Hospital, University of Pretoria, Hatfield 0028, South Africa; 2Department of Medical Microbiology, School of Laboratory Medicine & Medical Sciences, Medical School Campus, College of Health Sciences, University of KwaZulu-Natal-Natal, Durban 4041, South Africa; 3Department of Surgery, University of KwaZulu Natal, Congella, Durban 4013, South Africa; 4SAMRC Research Capacity Development Division, South African Medical Research Council, Tygerberg, Cape Town 4091, South Africa; 5School of Medicine, Department of Pathology & Laboratory Medicine, 1430 Tulane Ave., SL-79, New Orleans, LA 70112, USA; 6SAMRC Precision Oncology Research Unit (PORU), DSI/NRF SARChI Chair in Precision Oncology and Cancer Prevention (POCP), Pan African Cancer Research Institute (PACRI), University of Pretoria, Hatfield 0028, South Africa

**Keywords:** colorectal cancer, inflammation, metastasis, EMT, TLR, vimentin, HIV, NF-ĸB, inflammasome, epigenetics

## Abstract

**Simple Summary:**

Despite the reduced death rate that comes with improved HIV treatment, chances of developing cancers in HIV infected individuals remains high. The main culprit here is exaggerated inflammatory responses, referred to as chronic inflammation. Inflammation happens when the tissue is damaged by an outside (e.g., injury by falling) or internal (e.g., infection by bacteria/parasites/viruses) sources. Normally, inflammatory responses assist with fighting off infections and promoting wound healing. Opportunistic infections are common in HIV and constantly trigger chronic inflammation responses and ultimately cancer development. Cancers are dubbed the wound that does not heal because cancers survive better in a chronic inflammatory state. Several inflammatory pathways are known to promote colorectal cancer initiation. These can be common in both colorectal cancers and HIV. Toll like receptor inflammatory pathways are important in the detection of injuries. Their role and therapeutic endeavors in HIV and cancers are well studied. These pathways connect to specific inflammatory pathways which in return interconnects with several other pathways that are involved in HIV-related colorectal cancers. Their contribution in colorectal cancers is impactful hence the suggestion to target specific or best, interconnected inflammatory pathways with the hope of halting cancer initiation, development and progression.

**Abstract:**

The advancement of HIV treatment has led to increased life expectancy. However, people living with HIV (PLWH) are at a higher risk of developing colorectal cancers. Chronic inflammation has a key role in oncogenesis, affecting the initiation, promotion, transformation, and advancement of the disease. PLWH are prone to opportunistic infections that trigger inflammation. It has been documented that 15–20% of cancers are triggered by infections, and this percentage is expected to be increased in HIV co-infections. The incidence of parasitic infections such as helminths, with *Ascariasis* being the most common, is higher in HIV-infected individuals. Cancer cells and opportunistic infections drive a cascade of inflammatory responses which assist in evading immune surveillance, making them survive longer in the affected individuals. Their survival leads to a chronic inflammatory state which further increases the probability of oncogenesis. This review discusses the key inflammatory signaling pathways involved in disease pathogenesis in HIV-positive patients with colorectal cancers. The possibility of the involvement of co-infections in the advancement of the disease, along with highlights on signaling mechanisms that can potentially be utilized as therapeutic strategies to prevent oncogenesis or halt cancer progression, are addressed.

## 1. Introduction

Colorectal cancer (CRC) is the third most common and second most deadly cancer worldwide. The disease affects more men than women, with a lifetime risk of 1 in 23 and 1 in 25, respectively. More younger adults (<50 years old) are seen, with an incidence of a 1.5% increase every year from 2014 to 2018 [1,2]. By, 2040, the estimated number of new CRC cases is projected to be around 3.2 million and thus poses a great burden and continuing global public health challenge [2]. Immunosuppression is the main factor that predisposes people living with human immunodeficiency virus (HIV) (PLWH) to infection-related cancers. A study by O’Neill et al. [3] found the incidence of CRC to be the same in HIV and non-HIV-infected individuals. The study included 27 studies performed in North America (18), Europe (7), the Pacific region (4), and South America (1) [3]. However, this study did not include subjects from the low-middle income countries (LMIC), particularly those from the Sub-Saharan African regions. The challenge with LMIC is the disparities in socioeconomic status. Poor living conditions expose PLWH to opportunistic infections which are exacerbated by malnutrition. Poor sanitation can lead to exposure to fecal parasites including soil-transmitted helminths [4]. These parasites can cause tissue damage to the colorectum as they migrate through the body [5]. Other factors such as microbial infections and chemical agents can also cause injury, which then triggers inflammatory responses [6,7].

Prolonged exposure to inflammatory mediators such as cytokines, chemokines, and growth factors expressed during inflammatory responses result in a chronic inflammatory state associated with the initiation of CRC [8]. This has led to the use of nonsteroidal anti-inflammatory drugs (NSAIDs) for the treatment of CRCs [9]. These drugs showed an improvement and even regression in some patients, including those diagnosed with familial adenomatous polyposis (FAP), with an inherited mutation in the tumor suppressor gene, adenomatous polyposis coli (APC) [10]. Proinflammatory cytokines such as tumor necrosis factor (TNF) and interleukin (IL)-1 trigger the activation of inflammatory signaling pathways including nuclear factor kappa B (NF-ĸB), which is needed for the transcription of cytokine genes [11]. The activation of the NF-ĸB signaling pathway also promotes CRC progression [12]. Inflammatory cytokines promote the growth of tumor cells [13]. Tumor cells release cytokines that activate the NF-ĸB pathway, pushing tumor progression towards metastatic disease [14]. The activation of the NF-ĸB signaling pathway is initiated by the toll-like receptors (TLRs). The TLRs are expressed on the surface of colonic and innate immune cells. Upon activation by a pathogenic antigen(s), a cascade of signaling molecules is stimulated to elicit immunity against the pathogen and to initiate tissue repair mechanisms [15]. Both the TLRs activation and the subsequent activation of the NF-ĸB signaling pathways are associated with oncogenesis. The TLR-2, 4, and 9 have been specifically correlated with CRC [16,17,18]. TLRs are also expressed by tumor cells [19], which prompted investigations for the use of TLRs antagonists as a treatment of choice for CRC [20] and other cancers [21]. Patients with HIV were found to be at a higher risk of developing CRC [22]. Therapeutic interventions for HIV-1 using TLRs have long been suggested [23] and have been successfully shown to eliminate HIV-infected cells in pre-clinical studies [24]. This effect on inflammatory signaling pathways links CRC and HIV, which could perhaps be a strategy for “killing two birds with one stone”. This could therefore guide the efforts to develop therapeutic strategies for preventing CRC development and progression in PLWH who are on antiretroviral therapy. Herein, we discuss key inflammatory signaling pathways, with more focus on the TLRs- NF-ĸB/IFN and associated pathways and the possibility of targeting these pathways as a therapeutic strategy that could indiscriminately benefit HIV- and non-HIV-infected CRC patients. These therapies should take into consideration the effect of parasitic or microbial infections associated with CRC development as well.

## 2. HIV in Colorectal Cancer

PLWH present with CRC at a younger age than their HIV-negative counterparts. The median age at CRC diagnosis was 55 years (32–73 years), and the median period of HIV infection prior to CRC diagnosis was 15 years. An HIV-infected group had more smokers or people with a previous history of smoking than the negative group. Furthermore, high levels of tumor-infiltrating lymphocytes (TILs) and a tendency of presenting with right-sided CRC was reported [25]. In concurrence with this study, HIV-infected CRC patients presented at a much younger age than those without HIV. The recurrence rate was also reported to be higher in HIV-positive (14.7%) CRC than in HIV-negative (6.8%) patients [26]. This study also supports the reports indicating that PLWH have a fourfold increased risk of developing CRC [27]. One of the mechanisms for the increased risk may be due to the Transactivator of transcription (Tat) protein, which modulates HIV-1 gene expression and facilitates the efficacy of its viral transcription. Tat is known to have oncogenic properties, and this effect was used to investigate its role in CRC oncogenesis. The main factors associated with Tat-induced CRC were its ability to significantly induce cancer cells migration, thus facilitating cancer invasiveness and metastasis. Furthermore, Tat inhibits epithelial cytodifferentiation and apoptosis, both of which are the hallmarks of cancer [27]. 

Conventional CRC therapy compromises immune responses and predispose patients to opportunistic infections. The use of opportunistic infection prophylaxis along with HAART was shown to decrease the risk of getting such infections [28]. Recently, HIV therapy lamivudine (3TC) was shown to attenuate CRC progression [29]. Lamivudine is a nucleoside analog and reverse transcriptase inhibitor [30]. Retrotransposon insertions [31] and repeating units [32] are associated with CRC development and progression; hence, treatment with lamivudine could halt CRC progression more profoundly in p53-mutant cell lines. The mechanism of this drug includes the induction of the interferon (IFN) response gene and the DNA damage response [29]. Xu et al. (2020) found IFNγ to play an important role in the induction of the immune checkpoint (IC) by the CRC tissue microenvironment. Apart from inducing classical ICs such as programmed cell death protein 1 (PD-1) and programmed death-ligand 1 (PD-L1), the study identified three potential novel IC-related genes induced by IFNγ. The IFNγ-inducible lysosomal thiol reductase (IFI30), guanylate binding protein1 (GBP1), and guanylate binding protein 4 (GBP4) were found to be highly expressed in CRC tumors compared to controls. The study indicated that the efficacy of IC inhibitors as cancer immunotherapy in CRC might improve by the addition of more than one IC inhibitor, with a consideration of higher instead of lower IFNγ expression levels [33].

## 3. Cytokine Gene Transcription Signaling in Relation to HIV-Associated CRC

Evidently, controlling HIV replication appears to play a crucial role in attenuating CRC progression [34,35]. Thus, in the efforts of finding curative HIV therapy, combination therapy with TLR agonists and broadly neutralizing antibodies (bNAbs) successfully blocked HIV-1 replication but failed to do so when used individually in people who were on anti-retroviral (ARV) therapy. Preclinical studies indicate that HIV remission could be sustained without the use of ARVs [24]. The use of HAART cannot completely eradicate HIV because of the virus’s ability to hide in the immune cells and remain dormant (HIV latency) until such time that the treatment is ceased, allowing the virus to become active and replicate. Both cytokine IL-7 and C-C chemokine ligand (CCL) 19 could increase the rate of latent HIV infection threefold. In terms of productive HIV infection, IL-7 had a fivefold increase, whilst CCL19 had a twofold increase in productive infection [36]. Latency-reversing agents (LRA) could be used to reactivate HIV transcription in dormant cells, thus stimulating the expression of surface antigens that will trigger immune responses and concurrently prevent the release of virions by HAART. However, the treatment of HIV-infected peripheral mononuclear cells (PBMCs) with TLR agonists, gene expression modulator 91 (GEM91), and a gag antisense phosphorothioate oligonucleotide showed an increased viremia caused by the induction of TLR 9 stimulation through its CpG motif. This effect was demonstrated in dose-escalation studies [37] indicating that TLR signaling contributes to challenges related to HIV treatment and possibly CRC in PLWH. Although TLRs can be beneficial in inducing immune reactivation to clear infections and prevent cancer development and progression, in certain instances, TLRs promote cancer survival and metastasis. Thus, the TLR pathway for cancer therapy is dependent on the type of TLR and the type of cell producing it. Under normal cancer immunosurveillance, TLR agonists can be beneficial to cancer cells as well. Therefore, the design of TLR-targeting agonists as well as antagonists is a promising immunotherapeutic approach to cancer [38].

### TLR Signaling

Toll-like receptor family proteins are part of the innate immune system that initiates a cascade of inflammatory responses against microbial [39], parasitic [40], and viral [41] infections. There are 250 TLRs in total that have been identified thus far, with TLR 1–10 being identified in humans. The TLR family members are divided into vertebrate type (V-Type) and protostome type (P-Type). The V-Type are characterized by single-cysteine-cluster TLRs, while P-Type TLRs have several cysteine clusters. These TLRs can further be categorized into cellular (TLR 1, 2, 4, 5, 6, and 10) and intracellular membrane (TLR 3, 7, 8, and 9) TLRs (Figure 1). The TLRs are expressed on the surfaces of antigen-presenting cells (APCs) and recognize pathogen-associated molecular patterns (PAMPs) on various microorganisms. They are critical in sensing the microbiota and maintaining tolerance or triggering an immune response to pathogens. The TLRs, therefore, contribute to the prevention of gut microbial dysbiosis, associated with CRC initiation and progression [42]. 

## 4. Nuclear Factor Kappa B (NF-ĸB) Signaling

All TLRs subsequently end in the activation of NF-ĸB, which will guide the type of innate immune response elicited based on a specific pathogen [43]. Dysregulations in the NF-ĸB family of transcription factors are associated with inflammation in most cancers [44]. The NF-ĸB family consists of five connected components, the NF-κB1 and 2 (p50 and p52), RelA and B, and c-Rel [45]. These components form a homodimer or heterodimer transcriptional complex through the Rel homology domain (RHD). The RHD facilitates dimerization, binding to the promoter region of the DNA strand, and interaction with IκBs and RelA. The composition of the RelA subunit allows it to be the only member that can initiate DNA transcription. The inhibitor proteins IκBs interact with NF-ĸB as an on-and-off switch. This group consists of IκBα, IκBβ, IκBγ, IκBɛ, Bcl-3, IκBζ, and IκBNS. The Bcl-3, IκBζ, and IκBNS are responsible for transcriptional regulation within the nucleus, whilst IκBα, IκBβ, IκBγ, and IκBɛ are cytoplasmic inhibitors that assist in keeping NF-ĸB in its inactive form [46,47].

The IκBα binds as a monomer to other dimeric cytoplasmic inhibitors via their RHD blocking the DNA sequence needed for nuclear localization. Upon the activation of NF-ĸB by pathogen stimuli or other factors such as chemical agents, IκBα gets phosphorylated. However, the phosphorylation of IκBα is not sufficient for dissociation from the complex to allow for the mobilization of NF-ĸB into the nucleus to be initiated. Phosphorylation only marks IκBα for degradation by a specific ubiquitin-proteasome system that requires ATP. Complete degradations take place when sufficient ubiquitin multimers have been accumulated, and the phosphorylated IκBα polyubiquitin conjugates serve as a substrate for the 26S proteasome (Figure 2). This will then allow for the subsequent translocation of NF-ĸB into the nucleus [48]. 

A study by Jana et al. suggested that blocking NF-ĸB along with its activator, Activin, can reduce the chances of CRC metastasis, with the possibility of even halting cancer growth. Mechanistically, Activin-NF-ĸB induced an increase in the mouse double minute 2 homolog (MDM2) ubiquitin ligase and the subsequent degradation of p21 via phosphatidylinositol 3-kinase (PI3K) activation [49]. The p21, also known as cyclin-dependent kinase inhibitor 1, promotes cell cycle arrest, and its dysregulation is associated with oncogenesis. The maintenance of a homeostatic balance between cell proliferation and death is crucial for keeping cancer cells at bay. The p21 can serve as a tumor suppressor or oncogene. It can either promote or halt cancer proliferation by inhibiting or promoting apoptosis [50,51]. The progression of CRC in a metastatic stage was inhibited via the induction of lipocalin2 (LCN2) and the concomitant inhibition of the NF-κB/snail signaling pathway, which promotes EMT, migration, and invasion in CRC [52]. Treatment of colon cancer xenografts with prolyl hydroxylase 2 (PHD2) could inhibit metastasis through the inhibition of the NF-κB signaling pathway and the downregulation of EMT-related proteins including vimentin. Furthermore, PHD2 regulates inflammatory responses by the suppression of pro-tumorous immune cells such as myeloid-derived suppressor cells (MDSCs) and tumor-associated macrophages (TAMs). This induces the production of anti-tumor cytokines including G-CSF, TNF-α, IL-6, IL-8, IL-1β, and IL-4 [53].

### 4.1. Notch-Mediated NF-ĸB Inflammatory Response

It is important to note that the TLR4 signaling discussed earlier upregulates the activation of the Notch signaling pathway. Jagged1 (a ligand for Notch) is a direct TLR target gene that is overstimulated by IFN-γ [54]. In return, the overexpression of Notch1/2 suppresses the expression of TLR4-induced proinflammatory cytokines, TNF-α, and IL-6 and favor cancer-promoting anti-inflammatory cytokine IL-10. This results in an ultimate reduction in NF-κB transcription activity through extracellular signal-regulated kinase (ERK)-dependent signaling pathways [55]. The ERK fall under the mitogen-activated protein kinase (MAPK) family of signaling pathways, which are important in cell proliferation and differentiation. The dysregulation of this signaling cascade is implicated in oncogenesis [56]. Components of the NF-ĸB signaling pathway associate with the Notch signaling pathway, to promote inflammatory responses in cancer. The incorporation of Notch with TNF-α and κB kinase 2 (Ikk2) blocks anti-inflammatory machinery and promote oncogenesis [57] (Figure 3). Treatment with fructus aurantii extracts can also upregulate Notch/NF-κB/IL-1 signaling, leading to inhibition of colitis-associated CRC proliferation [58]. In CRC, active IKKα phosphorylates, silencing mediator of retinoic acid and thyroid hormone receptor (SMRT) along with specific Notch genes (*hes1, hes5, and herp2/hrt1*) facilitate its translocation into the nucleus, where cancer-favoring gene expression takes place. A compound referred to as BAY11-7082 dysregulates IKKα activity and significantly reduces cancer proliferation [59]. Interestingly, a study by Ahmed et al. [60] found that crosstalk between the Notch and NF-κB signaling pathways has a regulatory role in inflammatory responses and stem cell function during transmissible murine colonic hyperplasia (TMCH). Chronic administration of a Notch blocker dibenzazepine (DBZ) inhibited Notch and NF-κB signaling, resulting in the abrogation of hyperplasia. A significant expression of the cytokines IL-1α, granulocyte colony-stimulating factor (GCSF), monocyte chemoattractant protein 1 (MCP-1), macrophage inflammatory protein 2 (MIP-2), and keratinocyte-derived chemokine (KC) was observed. The authors noted that the Notch and NF-ĸB signaling pathways are associated with epithelial cancers and chronic intestinal inflammation, respectively. These pathways are commonly coactivated, making them a potential target for therapeutic endeavors [60].

Similarly, coactivation of the Notch and NF-ĸB signaling pathways plays a major role in HIV-related disease, including cancer. Notch4 was found to be responsible for tubulointerstitial injury and inflammation in HIV-associated nephropathy. The deletion of Notch4 resulted in a significant reduction in the inflammatory cytokines IL-6 and CCL2 and NF-κB transcription factors [61]. HIV-1 viral protein U (Vpu) was shown to inhibit the activation of NF-κB signaling to avoid anti-viral immune responses [62]. On the other hand, an HIV-1 accessory protein called viral protein R (Vpr) associates with transforming growth factor-β-activated kinase 1 (TAK1) to activate NF-κB and activator protein 1 (AP-1) signaling pathways, giving the virus access to host cell gene expression machinery [63]. AP-1 is responsible for the transcription of certain genes, including cytokines. One of the subcomponents of AP-1, c-jun gene, gets mutated during retroviral transduction and develops carcinogenic properties such as the ability to convert normal fibroblasts into cancer-associated fibroblasts (CAFs). Furthermore, mutated c-jun renders CAFs resistant to apoptosis by alkylating agents. The KRAS pathway is the major activator of the AP-1 pathway in CRC. The AP-1 signaling is activated through the induction of MAPK and the Wnt signaling pathway [64] (Figure 3). 

Other studies have also reported a crosstalk between the Wnt/β-catenin and NF-κB signaling pathways in inflammation. These pathways can both regulate the expression of genes responsible for cell differentiation, proliferation, and survival. Individually, NF-κB is known for its capacity to control inflammatory responses, whilst Wnt/β-catenin signaling is more important in tissue development and regeneration. These pathways have been shown to regulate each other via positive or negative feedback mechanisms depending on the type of the cell or tissue being targeted [65]. Several Wnt signal pathway-activated intestinal stem cells markers (Lgr5, EphB3, Ets2, Sox9 61, and Smoc2) have a significant role in promoting CRC metastasis. CRC cells with strong nuclear β-catenin localization are prone to undergoing epithelial mesenchymal transition (EMT) processes, thus initiating cancer invasiveness and metastasis. The EMT processes are responsible for the development of cancer stem cells in CRC and promote drug resistance [66]. 

### 4.2. Vimentin Gene Expression in HIV-Related CRC

The association between vimentin and cancer metastasis has been thoroughly studied and evaluated [67,68], leading to the acceptance and establishment of the use of vimentin as a marker for EMT-mediated metastasis [69,70]. High levels of vimentin were associated with poor prognosis and progression in CRC [71]. The immune expression of vimentin was associated with a high tumor grade in CRC. A significant correlation of vimentin and distant metastasis was also observed [72]. Blocking the NF-ĸB pathway was shown to inhibit the stemness, self-renewal, and migration capabilities of cancer cells [73]. Vimentin was identified as a regulator of NF-ĸB [74], suggesting that one of the mechanisms of vimentin-associated EMT is through the induction of the NF-ĸB signaling pathway (Figure 4). The correlation between high expression levels of vimentin, the NF-ĸB pathway, and related transcription factors is associated with poor prognosis in breast cancer patients [75,76]. Another study showed that NF-κB-mediated transforming growth factor-beta (TGF-β) induced the expression of vimentin in prostate cancer and was responsible for progression into metastatic disease [77]. A novel NF-κB-inducing kinase (NIK) and IκB kinase β (IKKβ) binding protein, referred to as NIBP, is required for the activation of the NF-κB alternative pathway. The upregulation of NIBP expression in colon cancer seemed to inhibit the expression of E-cadherin (Figure 4), whilst the expression of vimentin and CD44 remained high. The inhibition of the NF-κB pathway could not suppress the expression of CD44 and vimentin. The authors suggest that the inhibition of the NIBP might serve as an effective treatment for CRCs [78].

The NACHT, LRR, and PYD domains-containing protein 3 (NLRP3) inflammasome is responsible for defense against viral infections through the stimulation of IL-1β and IL-18 secretion [79]. Vimentin is required for the activation and regulation of the NLRP3. This concept was proven in mice studies in which wild-type mice succumbed to a systemic inflammatory response threefold more often than the vimentin-knockout mice [80]. The study indicates that vimentin is somewhat important in inflammatory responses known to influence oncogenesis. In a study by Yu et al., vimentin was shown to favor the secretion of a cancer-promoting cytokine, IL-10, through the stimulation of dendritic cells (DCs). Vimentin-stimulated DCs were also shown to have a decreased production of anti-cancer cytokines, IL-6 and IL-12 [81]. The expression of vimentin is itself regulated by interleukins. The expression of IL-2 was shown to induce vimentin 10 to 20-fold more, while IL-4 suppressed its expression by inhibiting IL-2-induced vimentin RNA [82]. The specific roles of these interleukins in CRC initiation, development, progression, and metastasis are reviewed in more detail by Li et al. [83].

In addition, vimentin serves as a substrate for retroviruses including HIV [84,85]. HIV-1 uses vimentin to alter the chromatin organization and distribution in infected cells, thus ascertaining vimentin’s contribution in HIV-1-associated cytopathogenesis and oncogenesis [86]. Vimentin was identified as a potential biomarker for HAART efficacy in PLWH [87] and a therapeutic target for HIV [88]. These findings provide a strong correlation between CRC-Vimentin-HIV and the potential use of vimentin in the effort to prevent and eradicate CRC in PLWH who are on long-term treatment with HAART. 

## 5. Interferon Signaling Pathways

Interferons are a family of inflammatory mediators secreted in response to various insults, including cancer cells and viruses. There are over twenty interferons identified thus far, categorized into three classes: IFN 1, 2, and 3 interferons. The IFN I are stimulated in response to viruses such as HIV. They primarily have a proinflammatory role, but prolonged exposure can lead to immune dysregulation. However, interferons have a contradictory role in cancer immunity [89] (Table 1).

In CRC, both IFN 1 and 2 have been found to have a dual effect on cancer development and progression. This can take place, in part, through the expression of immunosuppressive Tregs and MDSCs which inhibit the expression of IFN by T helper 1 cells, thus blocking anti-cancer immunity [99]. The presence of IFN within the tumor microenvironment facilitates anti-cancer mechanisms in several ways. These include promoting the expression of major histocompatibility complex (MHC) by cancer cells; enhancing T-cells and natural killer cells trafficking; stimulating the expression of anti-cancer cytokines and favoring T helper 1 polarization.. Dysregulation in the IFN signaling pathways makes an immense contribution to immunotherapy resistance, hence the consideration to include IFNs in the development of CRC therapies [100]. However, in the case of CRC in HIV infection, the timing of therapy in terms of chronic versus acute infection seems to be of great importance if one is to benefit from IFN therapy and still manage to circumvent the deleterious effects thereof [101].

The manipulation of the IFN signaling pathway via the administration of IFN-α2a could temporarily upregulate antiviral genes and prevent simian immunodeficiency virus (SIV) infection in rhesus macaques. Continued treatment resulted in IFN-1 exhaustion and increased infection, which was accompanied by a low CD4 count [102]. New HIV infections turn out to be more resistant to IFN than the chronic infections. Infections are considered chronic six months from the initial infection. The IFN resistance was attributed to HIV-1 replicative fitness rather than resistance to a specific type of IFN [103]. Contrary to these findings, the initial viral infections were resistant but, with time, became susceptible, specifically to IFN-α, with no difference in the replicative fitness of the HIV-1 infection [104]. 

### 5.1. Indoleamine 2,3 Dioxygenase 1 (IDO1)

IDO1 regulates T cells’ immune responses by degrading tryptophan in DCs and macrophages. During this process, T cells go through proliferation arrest, rendering them futile as a defense mechanism against cancer development [105]. The expression of IFN-γ induces the expression of both IDO1 (Figure 5) and the tryptophan selective transporter in the colon epithelial cell line [106]. Radiotherapy induces the expression of IDO1 via IFN 1 and 2 in CRC. As a result, the inhibition of IDO1 sensitizes CRC to radiotherapy and promotes radiation-induced apoptosis. Blocking IDO1-induced anti-cancer Th1 cytokines within the tumor microenvironment facilitates the overall eradication of cancer cells, even around the surrounding stroma. Consequently, a IDO1 blockade can serve as a protective mechanism against radiation-related adverse effects, particularly on the normal small intestinal epithelium [107]. The activity of IDO is reported to be high in PLWH that are on HIV treatment. This activity is observed from the initial stages of the infection and persists throughout the chronic stage. Furthermore, the IDO activity is associated with the size of the HIV reservoir, immune activation, and T cell exhaustion, thus indicating its role in immune-related metabolic aberrations in HIV persistence [108]. Because most cancers constitutively express IDO and its role in HIV has already been established, blocking IDO might be of great therapeutic importance, especially in PLWH. The inhibition of IDO restored the proliferation of tumor-specific T cells with less toxicity, indicating that IDO might be favorable more so in individuals who are already at risk of developing detrimental adverse effects, as in the case of HIV infection [105].

### 5.2. Interferon-Induced KRAS Signaling and Related Pathways

The Kirsten rat sarcoma virus (KRAS) gene provides instructions for making the KRAS protein which activates the intracellular transduction signaling pathway known as the RAS/MAPK pathway [109]. The KRAS mutations are the most common in cancers and affect 52% of CRCs. Oncogenic KRAS mutations mediate inflammation and promote immune evasion, leading to cancer initiation and progression. Proinflammatory cytokines such as IL-6 and 8 have been identified as transcriptional targets of the KRAS mutations which take place through the activation of the MAPK/PI3K signaling pathways. Inhibition of the MAPK pathway showed no clinical benefit in KRAS-mutant CRC, unless it was coupled with the suppression of IFN-stimulated gene (ISG) expression (Figure 5), leading to cancer cells apoptosis [110]. Further, KRAS mutations have been linked to the induction of PD-L1 expression, leading to immune evasion via the activation of the AKT/mTOR pathway [111]. Blocking Myc-inhibited KRAS^G12C^ mutation, leads to the induction of the IFN signaling pathway and the resultant reduction in tumor-favoring immunosuppressive cells and upregulation of anti-tumor mechanisms. Contrary to the finding of the previous study, Mugarza et al. [112] observed no significant effect in cold tumors when using combinatorial treatment with both anti-PD-L1 and KRAS^G12C^ inhibitors. Cetuximab is an anti-CRC drug that competitively inhibits binding of epidermal growth factor receptors (EGFR) to cancer cells thus preventing their proliferation and survival. Its efficacy is tested by the continual assessment of KRAS mutations. It has been determined that cetuximab is ineffective in metastatic CRC with KRAS mutations. A study found that one of the mechanisms of cetuximab is through the induction of the IFN/STAT1 signaling pathway, which promotes cancer survival rather than inhibition in tumor cells. This was due to cetuximab having a preference for IFN sensitive cells over tumor cells. The former cells upregulates IFN/STAT1 signaling and are resistant to treatment. The authors suggested that targeting the IFN/STAT1 signaling pathway, EGFR ligands, and related genes could serve as novel predictors of efficacy in KRAS mutations in cancer patients on cetuximab treatment [113]. 

The ArfGAP with dual PH domain-containing protein 1 (ADAP1) can upregulate certain T cell signaling programs in HIV-1 infection via the stimulation of KRAS/ERK/AP-1 signaling, leading to latent HIV-1 reactivation [114]. The study of the role of KRAS deficiency in CD8 T cell immune responses during acute viral infection indicated the importance of KRAS in the development and proliferation of both CD4 and CD8 T cells. KRAS was also found to modulate the TCR-induced activation of the Raf-1/MEK/ERK pathway in CD8 T cells [115]. All these are promising therapeutic approaches to deciphering the deleterious effects of KRAS mutations in HIV-related and metastatic CRC.

### 5.3. CRC and HIV-Associated Opportunistic Infections

Apart from viruses, other microbial organisms including bacteria and parasites have a way of manipulating the immune responses. Microbial dysbiosis in the gut induces CRC by modulating inflammatory responses, molecular networks, and the production of metabolites [116,117,118,119]. The microbiota composition in CRC is diverse and includes strains such as the *Bacteroides fragilis, Streptococcus gallolyticus, Enterococcus faecalis, Escherichia coli, Parvimonas, Peptostreptococcus, Prevotella, Alistipes, Akkermansia* spp., *Fusobacterium nucleatum, Porphyromondaceae, Coriobacteridae*, and *Methanobacterials* [118,120,121]. The abundance of the microbial species seems to differ according to the location, with the right-sided colon cancer predominately populated by *Firmicutes* and the sigmoid colon cancer populated by *Verrucomicrobia, Ruminococcaceae, Streptococcaceae, Clostridiaceae, Gemellaceae,* and *Desulfovibrio.* Similarly, the depletion of the gut bacteriome (*Akkermansia, Anaerovibrio, Bifidobacterium,* and *Clostridium*) was observed in chronically infected HIV-1 subjects. Depletion of these bacteriome is associated with metabolic disorders, CD8+ T cell responses, and chronic inflammation [122]. Another study showed the abundance of Proteobacteria—more predominantly, the Enterobacteriaceae family, which includes *Salmonella, Escherichia, Serratia, Shigella,* and *Klebsiella species*—in HIV. Additionally, *Staphylococcus, Pseudomonas,* and *Campylobacter* spp. are well studied opportunistic pathogens and sources of bacteremia in HIV-infected subjects [123].

Clay et al. identified inosine as a targetable microbial metabolite for 9potential therapeutic strategy for CRC antitumor immunity by looking at studies that correlated inosine with immune checkpoint inhibitors and the IFN-γ pathway (Figure 5) [121]. Moreover, microbiota can control the destiny of viral infections by the positive or negative regulation of the IFN signaling pathway. The eradication of intestinal and bile-derived microbiota can reduce the normal expression of IFN-λ, disrupting its antiviral activity. The same effect is observed with the impairment of the IFN-λ receptors [124]. The gut microbiota dysbiosis was also shown to influence the therapeutic response to HAART [125]. Targeting microbiota in HIV could decrease the chronic inflammation [126] related to CRC induction. 

## 6. HIV and Parasitic Co-Infections and CRC Oncogenesis

The connection between parasitic infections and oncogenesis is well studied and established. Parasites and cancers share the ability to manipulate and evade the immune system. These abilities ensure that cancers and parasites survive and multiply [127,128,129]. Protozoa and helminths are the major classes of parasites and could result in asymptomatic colonization or cause several clinical outcomes including life-threatening illness [130]. Some of the important cancer-associated parasitic infections are noted as helminths [131]. These parasites can cause intestinal obstruction that might require surgery and also lead to infections such as sepsis [132]. The chronic inflammation induced by these parasites is the major cause of oncogenesis [133]. A constrictive lesion at the sigmoid colon was observed in a patient with intestinal obstruction caused by *Schistosoma mansoni* infection. Upon further evaluation, the patient was found to have CRC, which could be a result of the chronic inflammation induced by the parasitic infection [134]. Moreover, CRC has been associated with a common intestinal protozoan, *Blastocystis* sp. The prevalence of *Blastocystis* sp. had a fivefold increase in stool samples of CRC subjects [135]. Other parasites such as *Cryptosporidium* [136], *Schistosoma japonicum* [137,138,139], *Schistosoma mekongi* [140], and intestinal helminths [141] have also been linked to CRC oncogenesis [131,136]. There are currently clinical trial studies underway aimed at validating the link between intestinal helminths and protozoa with CRC pathogenesis [142].

Several protozoans such as *Plasmodium, Entamoeba, Toxoplasma*, and *Leishmania* trigger host inflammatory signaling pathways by the secretion of cytokine molecules that are structurally similar to those of humans. The macrophage migration inhibitory factor (MIF) is a pleiotropic cytokine that induces acute inflammatory responses. The MIF binds to CD74 and induces the expression of TNF-α, IL-6, IL-8, and IL-12. These proteins have also been shown to activate the ERK1/2 and PI3K/Akt pathways [143], which are both implicated in oncogenesis [144] (vide supra). It is worth noting that the MIF can also promote the activation of the nucleotide-binding oligomerization domain-like receptor (NLR) pyrin domain-containing 3 (NRLP3) inflammasome [143], which is to be discussed in more detail in the next section.

## 7. Role of the Inflammasome Complex in HIV and CRC Oncogenesis

Inflammasome is one of the signaling pathways of the innate immune system that is induced by stimuli from microbial pathogens including viruses and the gut microbiota. The components of the inflammasome complex consist of the sensor proteins, NLRP 1-3, NLR family CARD domain-containing protein 4 (NLRC) 4, absent in melanoma 2 (AIM2), and pyrin. Caspase-1 is connected to the sensor proteins through the adaptor apoptosis-associated speck-like protein containing a caspase recruitment domain (ASC). ASC consists of the caspase activation and recruitment domain (CARD) and the pyrin domain of which the former is used to connect to the inflammasome sensor proteins. Cleavage of Caspase-1 is prompted by the connection of pro-caspase 1 to the complex through the CARD domain. This then leads to the activation of caspase-1-dependent pro-inflammatory cytokines, IL-1β and IL-18, and the induction of pyroptotic cell death. The ASC is important in the induction and facilitation of inflammatory responses [145,146].

Helminth infections can activate the NLRP3 inflammasome, resulting in the modulation of T helper 2 immunity [147]. The prevalence of HIV–helminths co-infections is high in the African population [148,149]. It is not clear whether helminths infections increase the susceptibility to HIV or whether HIV increases the susceptibility to helminths infections [150]. However, certain phenotypes of helminths suggest the former [151]. Helminth infections caused by *Schistosoma haematobium* and *Opisthorchis viverrine* have been shown to have carcinogenic properties. These parasites release metabolites such as oxysterols and catechol estrogens, which can integrate with the host cell DNA through the involvement of CYP450 enzymes. The ultimate formation of DNA-adducts is associated with helminth-associated oncogenesis [152]. Several cancers including colorectal hepatocellular carcinoma are associated with helminths infections [153]. Furthermore, activation of the NLRP3 inflammasome on its own is associated with cancers including the colon cancer. The NLRP3 complex can be activated via TLR- NF-κB signaling, which then induces the transcription and expression of the NLRP3 inflammasome cytokines, IL-1β, and IL-18. [154].

Dysregulation in the NLRP3-induced IL-1β and IL-18, which are critical for maintaining the integrity of the intestinal epithelial barrier, can result in CRC oncogenesis [155]. The level of IL-1β is upregulated in CRC, resulting in the activation of immunosuppressive cells such as MDSCs and promotion of cancer metastasis by facilitating the EMT processes. This cytokine can also induce angiogenesis and promote drug resistance in CRC. On the other hand, IL-18 has a pleiotropic effect, leading to both the suppression and promotion of CRC. IL-18 can suppress colitogenic microbiota associated with inflammasome-mediated intestinal dysbiosis but also promote endothelial cell stemness and angiogenesis [156]. Moreover, inflammasomes are the main inducers of inflammation, which promotes inflammatory diseases such as inflammatory bowel disease (IBD), which is associated with a high risk of CRC development [157]. A decrease in inflammasome-induced IL-18 can lead to the disruption of the IFN-γ pathway and ultimate microbial dysbiosis associated with CRC oncogenesis. Studies found these effects to be a result of a deficiency in inflammasome complex subunits including NLRP6. Taken together, the pro-inflammatory microenvironment generated by inflammasomes creates a suitable condition for CRC development and progression [155]. The most studied inflammasome, NLRP3, is the major culprit of chronic inflammation and microbial dysbiosis causing intestinal epithelial cells dysfunction and CRC oncogenesis; hence, there have been suggestions to use it as a potential therapeutic target for CRC [158]. NLRP3 was also shown to be increased in PLWH on HAART treatment. The inflammasome genes, NLRP3 and caspase-1 were found to be upregulated in immunological non-responders. The authors suggested that this could be the reason behind the constant immune activation that keeps these individuals immunocompromised [159]. The NLRP3 inflammasomehas a critical role in maintaining HIV-1 infection-associated inflammation and has been suggested as a therapeutic target for HIV as well. This serves as an indication that targeting NLRP3 might be efficacious in HIV-associated CRC [160].

## 8. Epigenetic Regulation of Inflammatory Signaling

The contribution of epigenetic dysregulations, such as microsatellite instability (MSI), caused by mutations in DNA mismatch repair genes in CRC is well established. These modifications happen in 10–15% of CRCs [161], whilst chromosomal instability (CIN) takes place in 65%–70% of sporadic CRCs [162], with approximately 65% of cases of CRCs being sporadic [163]. Epigenetic modifications are key regulators of inflammatory processes induced by inflammasomes, more so with the NLRP3 components. In preclinical studies, targeted therapeutic strategies for blocking NLRP3 inflammasome are performed using the diarylsulfonylurea-containing compound, MCC950. Alternatively, the inflammasome complex assembly and activation can be blocked with the use of NF-κB inhibitors, caspase-1 inhibitors, and the inhibition of the ATPase activity of the NLRP3 [164]. Recent advances have identified epigenetic mechanisms involved in the expression, assembly, and activation of the NLRP3 inflammasome as potential targets for therapeutic purposes [165]. However, when targeting the inflammasomes, caution should be taken to not disrupt their crucial role in fighting cancers and infections. Thus, a homeostatic balance ensuring the continual clearance of newly mutated cells and infections should be retained, and the epigenetic drugs developed should be precise and disease-specific. Microbiota can induce epigenetic modifications at the CpG islands in the promoter region of the NLRP3 gene [166]. This leads to the exaggerated inflammatory responses seen in inflammatory bowel disease and an increased risk of CRC oncogenesis [167,168]. Treatment with inhibitors such as JQ1 can protect the colon and inhibit NF-kB activation and the expression of NLRP3-associated components [169]. The activation of the NF-κB signaling pathway can cause DNA damage and induce epigenetic modifications such as CIN, which promotes oncogenesis [170,171]. This is attributed to the crosstalk between NF-κB and STAT3 signaling that contributes to challenges in CRC therapy. Further, NF-kB/IL-6 induces the expression of DNA (cytosine-5)-methyltransferase 1 (DNMT1), which downregulates the suppressor of cytokine signaling 3 (SOCS 3) needed for the regulation of cytokine-induced STAT3 signaling, leading to CRC oncogenesis [172]. Moreover, several microRNAs (miRNA) are involved in the regulation of the NF-κB signaling pathway and related transcription proteins in cancers. Some oncogenic viruses utilize miRNAs to activate NF-κB signaling and regulate immune responses to promote oncogenesis [173]. MicroRNA 146 targets and activates NF-κB, which in return upregulates miR-146 gene expression. To control its own expression, miR-146 downregulates the expression of interleukin-1 receptor-associated kinase 1 (IRAK1) and TNFR-associated factor 6 (TRAF6), which have a crucial role in the activation of NF-κB. The activation of NF-κB by miR-21 in some cancers is associated with therapeutic resistance and metastasis. The miR-301a is the most potent activator of NF-κB that negatively regulates it through its target gene, NF-κB repressing factor (NKRF). The upregulation of miR-301a in CRC is associated with increased cancer proliferation, migration, and invasion. This takes place through the activation of the TGF-β/Smad pathway, which downregulates the SOCS 6 and Smad4. MiR-301a also promotes cancer progression into the metastatic stage through the hyperactivation of the Wnt/β-catenin signaling. On the other hand, miR-30b blocks CRC progression by regulating the expression of KRAS, and it has been suggested as a prognostic indicator and therapeutic target of CRC [174].

## 9. Compounds/Agents Targeting Inflammatory Signaling in CRC

Wang et al. [175] have identified natural compounds that can potentially be used to target signaling pathways in CRC. These includes phytochemicals such as flavonoids and polyphenolic compounds such as epigallocatechin gallate (EGCG), which are targeted at the NF-κB signaling pathway. In addition to flavonoids, alkaloids and terpenoids can be used to target the Wnt/β-catenin signaling pathway. The most common cancer-related pathway, the MAPK/PI3K/Akt/mTOR signaling pathway, might benefit from treatment with polyphenolic compounds in addition to flavonoids [175]. Phytochemicals induce or inhibit cytochrome P450 enzymes [176] that are essential for mitigating cancer development, metastasis, and the response to anticancer therapies [177]. In addition to phytochemicals, nanoparticles have been used as targeted therapy against inflammatory signaling pathways in cancers. Nanoparticles serve as delivery systems of phytochemicals that modulate the function of specific inflammatory signaling pathways. These pathways include NF-κB, MAPK, PI3K/AKT/mTOR, and Wnt pathways discussed in this paper [178].

Several other agents and inhibitory drugs have been developed against the Wnt signaling pathway as a targeted therapy for CRC [179]. Targeted Wnt therapies are developed depending on the regulatory level and specific ligands. These include the extracellular and cell membrane levels, the cytoplasmic level, and the nuclear level. Lastly, the interrelated signaling crosstalk also has different levels that regulate the Wnt pathway [180]. NSAIDs are commonly used to relieve pain and reduce inflammation but have been shown to have chemo-preventative capabilities due to the association of chronic inflammation with oncogenesis [181,182]. The NSAIDs block cyclooxygenase-2 (COX-2) and reduce prostaglandin E2 synthesis associated with cancer proliferation and angiogenesis. The mechanisms of NSAIDs are also implicated in promoting cancer cells apoptosis [183,184]. This has led to the development of COX inhibitors for the treatment and prevention of cancers. The NSAID mechanism of action does not require COX for its chemo-preventative effect. NSAIDs can regulate several inflammatory pathways, including the NF-κB, the Wnt/β-Catenin, and the MAPK/Akt signaling pathways [185,186].

## 10. Conclusions

Improvements in CRC screening methods in Western/high-income countries allowed for much earlier detection, but the current treatment strategies are not beneficial to all individuals diagnosed with CRC. The introduction of HAART has come with the benefit of improving the lifespan of PLWH but with the disadvantage of rendering PLWH susceptible to opportunistic infections, microbial dysbiosis, and cancers, which are all associated with the induction of inflammatory responses. There are several signaling pathways associated with CRC, with the most important being those that regulate inflammatory responses. Herein, the review discussed specific inflammatory signaling pathways intertwined by different mechanisms and pathways. These includes but are not limited to microbial-regulated signaling pathways that contribute to CRC oncogenesis and progression. Cancer and HIV take advantage of these pathways, working in concert to ensure their survival and spreading. In this manner, the culprits always have an alternative loop way that they can fall into should their common pathway be blocked by certain inhibitors. Notably, in most cases, cancer/HIV would manipulate the mechanisms of a pathway that favors its initiation to interaction with a pathway that antagonizes it thus increasing its chances of survival and progression.

This could also be through the regulation of the expression of genes involved in anti-inflammatory signaling cascades. This includes related transcription factors involved in controlling and dampening anti-viral/tumor pro-inflammatory signaling pathways. Epigenetic modifications are well recognized in CRC, and these play a crucial role in regulating inflammatory mechanisms. Evidence shows that microbiota contribute to oncogenesis through the expression of biological molecules such as inflammasomes and short-chain fatty acids (SFCA). One of the mechanisms of SFCA action is the regulation of the histone deacetylases involved in inflammatory responses, cell proliferation, and migration. These biological molecules are also associated with CRC; however, this is beyond the scope of this paper and will not be discussed any further. These molecules trigger specific inflammatory signaling pathways which further contribute to the advancement of CRC. Previously, HIV was only associated with specific cancers such as Kaposi sarcoma, but over the years, its contribution to non-AIDS defining cancers is unquestionable. Thus, uncovering the complexity of several intertwined exits that HIV and related cancers manipulate to evade the immune system without severe adverse effects would be of great benefit, especially in aggressive and hard-to-treat cancers such as CRC. It is hoped that this approach can provide a fast and effective response that could also prevent metastasis and recurrence.

## Figures and Tables

**Figure 1 cancers-15-00748-f001:**
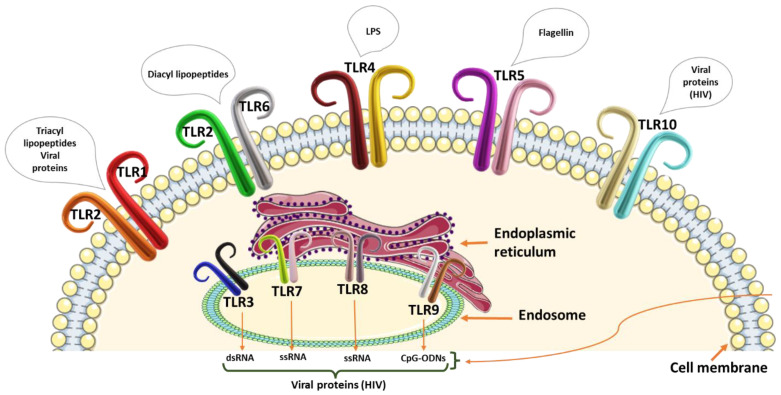
Representation of the human toll-like receptors on the cell membrane and the endosome with stimulatory pathogen-associated molecular patterns or danger-associated molecular patterns. Toll-like receptors 1 and 2 recognize triacyl lipopetides and viral proteins, whilst TLR2 and TLR6 recognize diacyl lipopeptides. The lipopolysaccharide (LPS) is recognized by TLR4 and flagellin by TLR5. TLR10 is specific for viral proteins including HIV protein receptors. The endosome TLR 3 recognizes double-stranded RNA (dsRNA). TLR 7 and 8 recognize single-stranded RNA (ssRNA) molecules, and TLR9 recognizes unmethylated CpG motifs. All endosome TLRs have the ability to recognize viral proteins.

**Figure 2 cancers-15-00748-f002:**
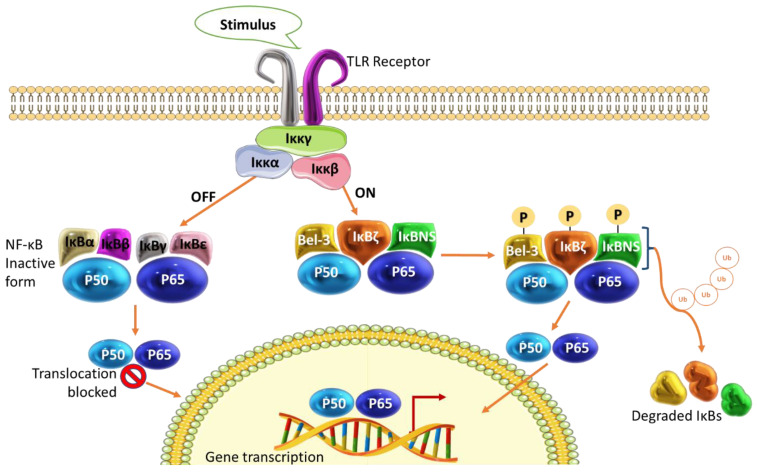
Representation of the NF-κB signaling pathway. The IκBα, IκBβ, IκBγ, and IκBɛ inactivate and prevent NF-ĸB from translocating into the nucleus, thus switching off the transcription of inflammatory genes. The Bcl-3, IκBζ, and IκBNS are mainly associated with facilitating transcription within the nucleus. The phosphorylation of these IκBs marks them for degradation by ubiquitinase, which then allows NF-ĸB to be translocated into the nucleus for the induction of cytokine gene transcription.

**Figure 3 cancers-15-00748-f003:**
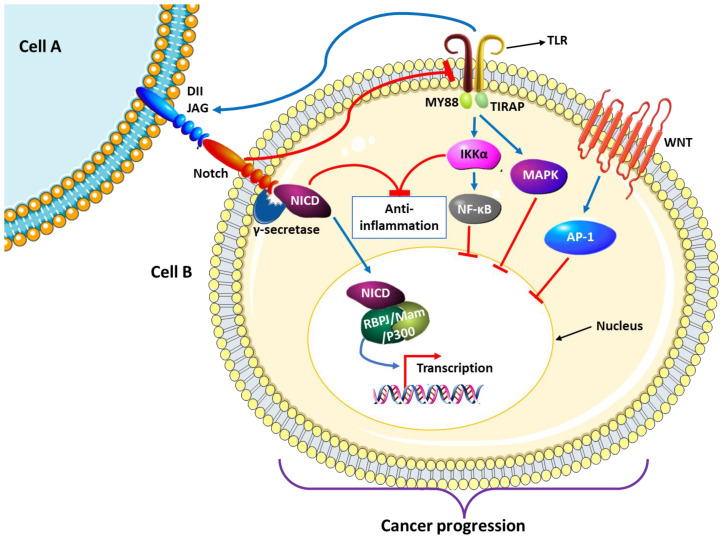
Notch works in synchrony with TLR-NF-κB signaling to control inflammatory responses that are central to cancer development. The TLRs can activate Notch via jagged proteins, and in return, the overexpression of Notch negatively regulates TLRs, resulting in a reduction in NF-κB signaling transcription activity, thus favoring cancer progression. The synergy between the components of the Notch and the NF-κB signaling leads to cancer progression by blocking anti-inflammatory protein expression. Mutations of the components of the AP-1 pathway promote cancer progression during viral infections. This pathway can be activated through the Wnt pathway, known for its ability to promote CRC progression.

**Figure 4 cancers-15-00748-f004:**
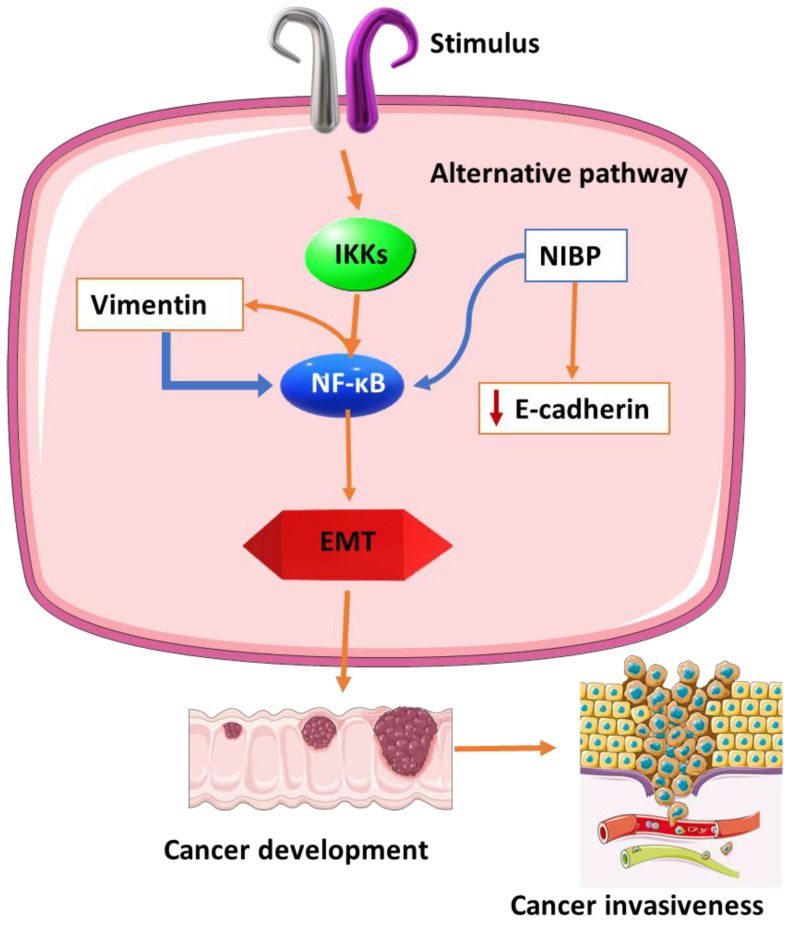
Vimentin as a regulator of the NF-κB pathway. Vimentin induces the NF-ĸB pathway to promote the stemness, self-renewal, and migration ability of cancer cells (EMT). In EMT, cells lose their adhesiveness and gain migratory and invasive capabilities. The NF-ĸB pathway can also induce the expression of vimentin, thus doubling up in promoting EMT in cancers. The alternative NF-κB-inducing kinase (NIK) and IκB kinase β (IKKβ) binding protein (NIBP) inhibits the expression of one of the EMT markers (E-cadherin) but fails to inhibit the expression of vimentin. EMT—epithelial–mesenchymal transition.

**Figure 5 cancers-15-00748-f005:**
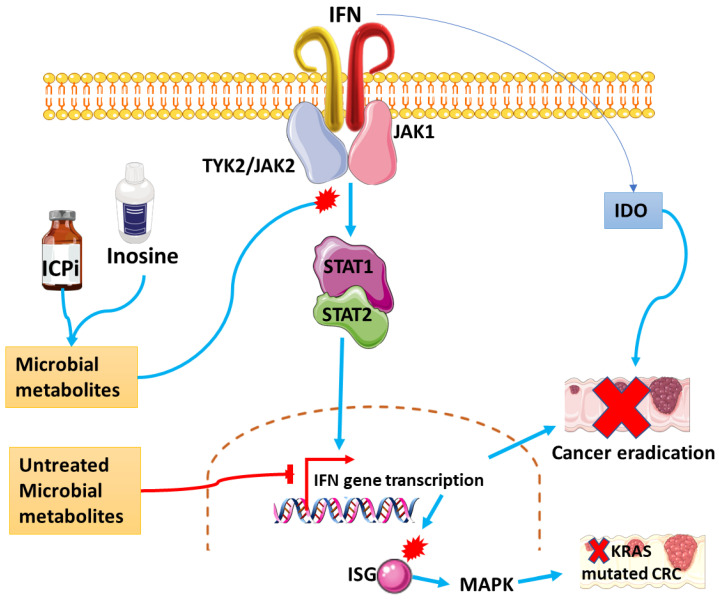
Interferon and related pathways involved in the regulation of HIV-related CRC mechanisms. The IFN-induced activation of the TLRs leads to downstream signaling which result in the expression of IFN genes and inflammatory cytokines. Microbial dysbiosis can regulate the expression of IFN cytokines to promote oncogenesis. The combination of inosine and ICPi can be used as a potential targeted therapy for microbial metabolites, thus blocking processes that lead to the induction of CRC. KRAS mutations are known for their ability to promote CRC induction. Treatment of CRC with KRAS mutations requires inhibition of both the MAPK and IFN signaling to halt cancer progression hence the encouragement to target interconnected signaling pathways Components of the IFN pathway, along with IDO, can sensitize cancer cells to radiation therapy and assist in the eradication of the disease. ICPi—Immune checkpoint inhibitor, ISP- Interferon stimulated gene.

**Table 1 cancers-15-00748-t001:** Interferons and their contribution towards HIV-related CRC.

Subgroups	Classes	Receptors	Role in HIV-Related CRC	References
IFN I	α β κ ε ω δ τ	IFNAR1/2	Pro-cancerous Promote immune evasion Susceptible to genetic mutations Promote chronic inflammation Facilitate cancer invasion and metastasis via EMT Upregulate PDL-1- and IDO1-promoting metastasis Regulatory T cells (Tregs) activation, thus suppressing CD8 T cells anti-cancer mechanisms Anti-cancerous Inhibit angiogenesis Facilitate anti-cancer immune mechanisms Cancer cell cycle arrest Inhibit cancer cell proliferation and promote apoptosis Potential therapeutic efficacy in combinatorial therapies	[89,90,91]
IFN II	γ	IFNGR1/2	Pro-cancerous IL-10 superfamily of cytokines (has cancer-promoting abilities) Susceptible to genetic mutations Upregulate PDL-1- and IDO1-promoting metastasis Tregs activation, thus suppressing CD8 T cells anti-cancer mechanisms Anti-cancerous Cancer cell cycle arrest Facilitate natural killer cell activity Facilitate antigen presentation and lysosome activity of macrophages Increase specificity of MHC class I and II molecules	[90,92]
IFN III	λ1 λ2 λ3 λ4	IFN-λ R1/2	Pro-cancerous Expression significantly correlated with immune checkpoints (CTLA4, PD1, PDL1, Tim3, IDO1, and LAG3) and poor survival Gene mutations render susceptibility to CRC initiation and development Anti-cancerous Regulate cancer cells apoptosis through IL-15 Potential effective therapeutic target Inhibits macrophage HIV infection, thus limiting the effects of HIV on CRC Upregulates Th1 and downregulates Th2 response during infection	[33,93,94,95,96,97,98]

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
