# Peer review of "Unraveling the Complex Interconnection between Specific Inflammatory Signaling Pathways and Mechanisms Involved in HIV-Associated Colorectal Oncogenesis"

_cancers, 2023, doi:10.3390/cancers15030748_

Round 1

Reviewer 1 Report

The authors reviewed literature on "Unraveling the Complex Interconnection Between Specific Inflammatory Signaling Pathways and Mechanisms Involved in HIV-Associated Colorectal Oncogenesis". Cross disease link could ease management of the diseases. This is an excellent piece of writing. However, this paper needs to cite references appropriate place. Here are some examples:

Line 37-38: Need appropriate citations. GLOBOCAN 2020 is the original source of information.

Line 44: Need appropriate citation after mentioned author name.

Line 52-53: Need appropriate citation too.

Line 86: “People living with HIV” already abbreviated as PLWH. So be consistent throughout the manuscript and follow the same for other abbreviated terms.

Line 94: Need appropriate citation

 Always remember to cite original source after each specific information even though you need to repeat same source reference. Follow these instructions throughout the manuscript.

Write the objective more precisely and add the pathways information what you have discussed in the following segments. 

This report lacks a link of parasites with HIV and CRC. Discuss it comprehensively. It would be better if you discussed this point under a subheading. 

This report focused on the underlying mechanisms of HIV patients with CRC. There is no specific discussion on the treatment/management. Either it can be added to this report, or you can remove any suggestion of treatment without providing a specific reason. 

Revise the conclusion and make concise. The conclusion is not aligned properly with the objective of this study. 

Author Response

Dear Reviewer,

Kindly receive the attached response letter.

Thank you for taking the time to review this manuscript.

Reviewer 2 Report

I am working for patients with colorectal cancer, and I have never read such an interesting report about CRC therapy and HIV therapy.

Your report had statements which was very hard to understand, especially the dyscription about pathways of cell communication signals. And there are some points of which I can't recognize the meaning,

1 Is there any relation between HIV and other cancer than CRC, for example, esophagus, lung, or head and neck cancer?

2 There are some statements with { } in line471, 472 and others. Are they references?

Author Response

(The authors gave the same response as above.)

Round 2

Reviewer 1 Report

Authors revised manuscript sufficiently and can be accepted.